# Ranked severe maternal morbidity index for population-level surveillance at delivery hospitalization based on hospital discharge data

Elena V. Kuklina[1], Alexander C. Ewing[1], Glen A. Satten[1], William M. Callaghan[1], David A. Goodman[1], Cynthia D. Ferre[1], Jean Y. Ko[1,2], Lindsay S. Womack[1,2]*, Romeo R. Galang[1], Charlan D. Kroelinger[1]

1 Division of Reproductive Health, National Center for Chronic Disease Prevention and Health Promotion, Centers for Disease Control and Prevention, Atlanta, Georgia, United States of America, 2 United States Public Health Service, Commissioned Corps, Rockville, Maryland, United States of America

* LWomack@cdc.gov

**Data Availability Statement:** The data underlying the results presented in the study are public use and available for purchase from the Healthcare

## Abstract

### Background

Severe maternal morbidity (SMM) is broadly defined as an unexpected and potentially life-threatening event associated with labor and delivery. The Centers for Disease Control and Prevention (CDC) produced 21 different indicators based on International Classification of Diseases, 9th Revision, Clinical Modification (ICD-9-CM) hospital diagnostic and procedure codes to identify cases of SMM.

### Objectives

To examine existing SMM indicators and determine which indicators identified the most in-hospital mortality at delivery hospitalization.

### Methods

Data from the 1993–2015 and 2017–2019 Healthcare Cost and Utilization Project's National Inpatient Sample were used to report SMM indicator-specific prevalences, in-hospital mortality rates, and population attributable fractions (PAF) of mortality. We hierarchically ranked indicators by their overall PAF of in-hospital mortality. Predictive modeling determined if SMM prevalence remained comparable after transition to ICD-10-CM coding.

### Results

The study population consisted of 18,198,934 hospitalizations representing 87,864,173 US delivery hospitalizations. The 15 top ranked indicators identified 80% of in-hospital mortality; the proportion identified by the remaining indicators was negligible (2%). The top 15 indicators were: restoration of cardiac rhythm; cardiac arrest; mechanical ventilation; tracheostomy; amniotic fluid embolism; aneurysm; acute respiratory distress syndrome; acute

Cost and Utilization Project (HCUP) - National (Nationwide) Inpatient Sample (NIS) via their website https://www.hcup-us.ahrq.gov/nisoverview.jsp. All interested researchers can access the data through HCUP directly. We are not permitted to share the data or make it available as per the data use agreement with HCUP. The data use agreement required of all users is located here: https://hcup-us.ahrq.gov/team/NationwideDUA.jsp. The authors did not have any special access privileges to these data.

**Funding:** The author(s) received no specific funding for this work.

**Competing interests:** The authors have declared that no competing interests exist.

myocardial infarction; shock; thromboembolism, pulmonary embolism; cerebrovascular disorders; sepsis; both DIC and blood transfusion; acute renal failure; and hysterectomy. The overall prevalence of the top 15 ranked SMM indicators (~22,000 SMM cases per year) was comparable after transition to ICD-10-CM coding.

## Conclusions

We determined the 15 indicators that identified the most in-hospital mortality at delivery hospitalization in the US. Continued testing of SMM indicators can improve measurement and surveillance of the most severe maternal complications at the population level.

## Introduction

Severe maternal morbidity (SMM) is broadly defined as an unexpected and potentially life-threatening event associated with labor and delivery [1]. SMM is associated with prolonged [2] and increased costs of hospital stay [3], fetal deaths [4], preterm deliveries [5], and long-term consequences for maternal physical and mental health [1, 6, 7]. At the population level, SMM surveillance is commonly conducted using administrative hospital discharge data based on International Classification of Diseases (ICD) codes [8]. The Centers for Disease Control and Prevention (CDC) produced an index consisting of 21 indicators based on ICD diagnostic and procedure codes to identify SMM cases [9]. Combining the 21 indicators into a single index enables population-level surveillance of life-threatening events, interventions, and organ failure types given analyzing individual indicators may not be feasible due to rare occurrence at delivery hospitalization. Research shows combining multiple indicators into a single index offers consistency in monitoring trends and is more straightforward to both communicate and understand than are individual indicators described in complex medical terminology [10].

However, some individual SMM indicators have measurement concerns or may lack specificity for disease severity. Numerous studies have assessed measurement issues for individual SMM indicators (e.g., blood transfusion, disseminated intravascular coagulation [DIC], eclampsia) [11–13]. For example, blood transfusion and DIC are the most prevalent SMM indicators in the US [9], yet these indicators have low sensitivity (65% and 22%, respectively) [13] in validation studies using medical records and a substantial proportion of hospital discharge records with blood transfusion and DIC do not reflect severe morbidity [11, 13]. Similarly, a previous study has reported a low positive predictive value (41.7%) for ICD codes associated with eclampsia [14]. Composite measurements, like the SMM index, only perform well if individual indicators in the index meet specific criteria (reliability, validity, feasibility) [10].

To determine how well individual indicators perform in identifying severe morbidity, our study ranks individual indicators according to how they identify in-hospital mortality, a well-accepted measure of severity [2]. We rank each indicator hierarchically according to its contribution to the overall SMM index using its population attributable fraction (PAF) of in-hospital mortality and a novel signal-to-noise ratio (SNR), that compares the level of a signal (measurement) to the level of background noise (variability) [15]. To assess whether patterns and prevalence of SMM indices remained consistent across the transition from ICD-9 to ICD-10 coding, we estimated predicted prevalence for 2017–2019 using ICD-9 coded data and compared these to the observed prevalence using ICD-10 coded data.

## Methods

We used the 1993–2019 National Inpatient Sample (NIS), the largest database of all-payer hospital discharge records in the US [16]. The NIS is a uniform, standardized dataset based on state in-patient data that are contributed to the Healthcare Cost and Utilization Project that is weighted by the Agency for Healthcare Research and Quality (AHRQ) to produce nationally representative estimates.

The NIS contains up to 40 diagnosis and 25 procedure codes per discharge including Diagnosis-Related Group (DRG) codes, as well as hospitalization characteristics, including patient demographics, admission source, discharge disposition, length of stay, and hospital charges. In-hospital mortality was identified using the variable "died during hospitalization." To ensure comparability of data across all years, AHRQ created trend weights for 1993–2011 that are consistent with the 2012 and later redesigned NIS datasets [17]. The cross-sectional nature of NIS prevented identification of multiple delivery hospitalizations for the same woman; thus, the unit of analysis in this study was the delivery hospitalization, not the individual.

Due to the transition from ICD-9 to ICD-10 coding on October 2015, we first limited our analysis to delivery hospitalizations between January 1993 and September 2015 for comparability of codes across years and to obtain a sufficient sample size to produce reliable estimates of SMM and death, since these are rare events. We identified delivery hospitalizations using a previously described algorithm [18].

### SMM indicators

The starting point for indicator selection was the 21 current SMM indicators identified by CDC; however, as discussed previously, there are measurement concerns for DIC, blood transfusion, and eclampsia [11]. We recoded these three indicators having measurement concerns. We recoded the data into three mutually exclusive indicators for DIC and blood transfusion: 1) a combination indicator, including codes for both DIC and blood transfusion; 2) an indicator including DIC codes but no blood transfusion codes; and 3) an indicator including blood transfusion codes but no DIC codes. The eclampsia indicator was recoded into a new indicator for severe types of hypertensive disorders of pregnancy and included severe pre-eclampsia, HELLP syndrome (hemolysis, elevated liver enzymes, and low platelet count), or eclampsia, consistent with the latest revision of the Canadian SMM index [2]. In summary, we removed the three current indicators with measurement issues (DIC; blood transfusion; eclampsia), and added four recoded indicators (DIC and blood transfusion; DIC without transfusion; transfusion only; severe pre-eclampsia, HELLP syndrome, or eclampsia), resulting in 22 SMM indicators for ranking according to how each indicator identifies in-hospital mortality.

### Statistical analysis

We calculated the prevalence, in-hospital mortality rate (%) and relative risk (RR) for each of the 25 SMM indicators (21 current indicators and four recoded indicators). Next, we calculated the PAF for the in-hospital mortality rate of each indicator [19]. PAFs for all SMM indicators were calculated using the formula $p_d ((RR-1)/RR)$ where $p_d$ is the proportion with the indicator among all hospital discharge records with in-hospital death and RR is the estimated relative risk of indicator-specific in-hospital death [19]. The estimated RR associated with a specific indicator was obtained from log-linear models, adjusting for year, payer, age, race [20], hospital region, hospital location (urban, rural and teaching), and all other SMM indicators. Missing was treated as a separate category when analyzing data on race. Covariates were selected a priori, based on the literature and clinical significance. Some indicators had a large PAF due to high prevalence, but a low in-hospital mortality rate indicating these indicators

may not represent severe complications. To address this, we developed a SNR [15] formula: the PAF for an indicator (signal) divided by the increase in standard deviation (noise) in the entire ranked SMM index that occurs when the indicator is included in the model compared to when it is not included (indicator-specific SNR equation, S1 Appendix). A higher SNR value suggests that an indicator's higher PAF is due to a higher mortality rate rather than a higher prevalence in the population.

We used a hierarchical, iterative process to rank the 22 indicators (21 current indicators, removing three indicators with measurement issues and adding four recoded indicators), ordering by the highest SNR. This process included: 1) calculating the SNR for each indicator and ranking the indicators by this value; 2) selecting the indicator with the highest SNR; 3) removing all as-yet-unselected indicators from a delivery hospitalization if that hospitalization also had the selected indicator. This final step in an iteration forces the selected indicator to be mutually exclusive of the remaining indicators, regardless of whether the hospitalization included other co-occurring indicators. This step is appropriate for our application, as we only wanted to rank indicators by their capacity to identify deaths not already identified by indicators having higher ranking. We repeated this iterative, ranking process until all 22 indicators were selected and mutually exclusive.

When an indicator was selected and made mutually exclusive, the resulting PAF was recalculated and designated as the *hierarchical* PAF (hPAF) and was held constant for subsequent iterations. Because the indicator conditions were made mutually exclusive, the cumulative hPAF comprised of the ranked indicators was then calculated by summing the hPAF values. This approach to indicator selection allowed us to rank the indicators in order of importance in identifying in-hospital mortality.

We then calculated the annual SMM prevalence per 10,000 delivery hospitalizations based on (1) all 21 current SMM indicators identified by CDC, (2) 20 current SMM indicators (excluding blood transfusion), (3) blood transfusion alone, and (4) the top 15 ranked SMM indicators that identified the most in-hospital mortality. To assess whether patterns and prevalence using the top ranked SMM indicators remained consistent across the ICD coding transition, we estimated predicted prevalences for 2017–2019 using ICD-9 coded data and compared these to the observed prevalence calculated using ICD-10 coded data. The predicted estimates and 95% confidence intervals were based on logistic regression models using NIS data from 2010–2015, preceding the ICD coding transition, with the year as a covariate and the prevalence of the individual top ranked SMM indicators as outcomes. We excluded data from 2016 as this was the first year after the ICD transition and coding practices were in transition. Percent differences were calculated to compare predicted values with observed values.

All national estimates and 95% confidence intervals were produced using SUDAAN 11.0.3, using NIS-provided weights and design variables. All other analyses were conducted using SAS 9.4 (SAS Institute Inc).

### Ethics approval

Because the NIS are de-identified data, this study did not require approval by an institutional review board at the CDC.

### Results

The study population consisted of 18,198,934 hospitalizations representing 87,864,173 US delivery hospitalizations of women ages 12–55 years (1993–2015; S1 Table). There were 0.8 in-hospital deaths per 10,000 delivery hospitalizations (n = 6,685). The indicator-specific prevalence, in-hospital mortality rate, PAF, and SNR for the 21 current SMM indicators and the

**Table 1. Prevalence, in-hospital mortality, population attributable fraction (PAF), and signal-to-noise ratio (SNR) for severe maternal morbidity indicators (SMM), Nationwide Inpatient Sample, 1993–2015 (weighted sample = 87,864,173, unweighted sample = 18,198,934).**

| SMM Indicators | Prevalence among delivery hospitalizations | | In-hospital mortality | | PAF[a] | SNR[b] |
|---|---|---|---|---|---|---|
| | n[c] | Per 10,000 deliveries (95% CI[d]) | n | % (95% CI[d]) | % | |
| Severe pre-eclampsia, HELLP[d] syndrome, or eclampsia[e] | 1,414,150 | 160.9 (155.8, 165.9) | 1,379 | 0.1 (0.1, 0.1) | 9.8 | 0.8 |
| Blood transfusion | 615,984 | 70.1 (67.7, 72.5) | 2,143 | 0.3 (0.3, 0.4) | 21.4 | 2.6 |
| Blood transfusion without DIC[d,e] | 576,929 | 65.6 (63.4, 67.9) | 1,065 | 0.2 (0.2, 0.2) | 13.9 | 1.7 |
| DIC | 209,700 | 23.9 (23.0, 24.8) | 2,029 | 1.0 (0.9, 1.1) | 16.6 | 3.4 |
| DIC[d] without blood transfusion[e] | 170,644 | 19.4 (18.6, 20.3) | 951 | 0.6 (0.5, 0.6) | 12.2 | 2.8 |
| Hysterectomy | 73,178 | 8.3 (8.1, 8.6) | 780 | 1.1 (0.9, 1.2) | 3.9 | 1.3 |
| Eclampsia | 72,025 | 8.2 (7.9, 8.5) | 230 | 0.3 (0.2, 0.4) | 0.1 | 0.0 |
| Acute respiratory distress syndrome | 46,070 | 5.2 (5.1, 5.4) | 2,401 | 5.2 (4.7, 5.7) | 12.0 | 5.2 |
| Pulmonary edema/acute congestive heart failure | 42,256 | 4.8 (4.6, 5.0) | 372 | 0.9 (0.7, 1.1) | -0.6 | -0.3 |
| Mechanical ventilation | 40,320 | 4.6 (4.4, 4.7) | 3,351 | 8.3 (7.7, 9.0) | 49.2 | 23.0 |
| Both DIC[d] & blood transfusion[e] | 39,056 | 4.4 (4.2, 4.7) | 1078 | 2.8 (2.4, 3.1) | 14.0 | 6.6 |
| Acute renal failure | 35,409 | 4.0 (3.9, 4.2) | 1,212 | 3.4 (3.0, 3.9) | 4.2 | 2.1 |
| Sepsis | 33,403 | 3.8 (3.6, 4.0) | 976 | 2.9 (2.5, 3.4) | 9.2 | 4.7 |
| Cerebrovascular disorders | 28,715 | 3.3 (3.2, 3.4) | 968 | 3.4 (2.9, 3.9) | 12.2 | 6.8 |
| Severe anesthetic complications | 22,079 | 2.5 (2.4, 2.7) | 214 | 1.0 (0.7, 1.3) | 1.8 | 1.1 |
| Shock | 20,290 | 2.3 (2.2, 2.4) | 1,027 | 5.1 (4.4, 5.7) | 1.4 | 0.9 |
| Thromboembolism, pulmonary embolism | 13,866 | 1.6 (1.5, 1.7) | 528 | 3.8 (3.1, 4.6) | 5.0 | 4.0 |
| Heart failure during surgery or procedure | 10,155 | 1.2 (1.1, 1.2) | 135 | 1.3 (0.8, 1.8) | -0.2 | -0.2 |
| Sickle cell crisis | 8,360 | 1.0 (0.9, 1.0) | 68 | 0.8 (0.4, 1.2) | 0.7 | 0.7 |
| Cardiac arrest | 5,347 | 0.6 (0.6, 0.7) | 1,952 | 36.5 (33.6, 39.5) | 25.0 | 32.1 |
| Restoration of cardiac rhythm | 5,220 | 0.6 (0.6, 0.6) | 1,932 | 37.1 (34.0, 40.1) | 25.2 | 32.6 |
| Amniotic fluid embolism | 3,858 | 0.4 (0.4, 0.5) | 612 | 15.9 (13.3, 18.5) | 3.1 | 4.6 |
| Tracheostomy | 2,104 | 0.2 (0.2, 0.3) | 260 | 12.3 (9.3, 15.4) | -2.6 | -5.3 |
| Acute myocardial infarction | 1,633 | 0.2 (0.2, 0.2) | 134 | 8.2 (5.2, 11.2) | 0.2 | 0.4 |
| Aneurysm | 1,077 | 0.1 (0.1, 0.1) | 51 | 4.8 (2.0, 7.5) | 0.8 | 2.2 |

[a] Population attributable fraction (PAF) calculated adjusting for other SMM indicators in this table. PAFs for all SMM indicators were calculated using the formula $p_d$ ((RR-1)/RR) where $p_d$ is the proportion with the indicator among all hospital discharge records with in-hospital death and RR is the estimated relative risk of indicator-specific in-hospital death [19]. The estimated RR associated with a specific indicator was obtained from log-linear models, adjusting for year, payer, age, race, hospital region, hospital location (urban, rural and teaching), and all other SMM indicators.

[b] Signal-to-noise ratio (SNR) is the selection criterion used to select the indicator that would maximize the ratio of the expected number of SMM cases detected (signal) to the increase in the standard deviation of the cumulative sum of indicators (noise).

[c] Weighted count (n).

[d] Other abbreviations: 95% CI: 95% confidence interval. HELLP syndrome: hemolysis, elevated liver enzymes, and low platelet count. DIC: disseminated intravascular coagulation.

[e] Recoded indicator

NOTE: The total number of indicators presented in this table is 25. This includes the 21 current CDC SMM indicators and four recoded indicators.

four recoded indicators (n = 25 indicators) are summarized in Table 1. The recoded indicator for severe pre-eclampsia, HELLP syndrome, or eclampsia was most common (160.9 per 10,000 delivery hospitalizations). However, its in-hospital mortality rate (0.1%) was the lowest of all indicators and 10 other indicators had a higher PAF (>9.8). Blood transfusion and the recoded blood transfusion without DIC were the second and third most common indicators at 70.1 and 65.6 per 10,000 delivery hospitalizations, respectively, but had low mortality rates (0.3% and 0.2%). Indicator-specific mortality rates were greatest for restoration of cardiac rhythm (37.1%), cardiac arrest (36.5%), and amniotic fluid embolism (15.9%).

**Table 2. Results of the hierarchical population attributable fraction (hPAF) approach for severe maternal morbidity indicator selection, with the selection iteration (k) of each indicator, the cumulative hPAF, and the signal-to-noise ratio (SNR) selection criterion at iteration of selection.**

| Indicator[a] | k[b] | hPAF[c] | Cumulative hPAF[d] | SNR[e] |
|---|---|---|---|---|
| | | % | % | |
| Restoration of cardiac rhythm | 1 | 25.2 | 25.2 | 32.6 |
| Cardiac arrest | 2 | 14.4 | 39.5 | 69.4 |
| Mechanical ventilation | 3 | 29.4 | 69.0 | 23.0 |
| Tracheostomy | 4 | 0.6 | 69.6 | 51.0 |
| Amniotic fluid embolism | 5 | 1.4 | 70.9 | 28.7 |
| Aneurysm | 6 | 0.3 | 71.2 | 13.3 |
| Acute respiratory distress syndrome | 7 | 2.3 | 73.6 | 5.3 |
| Acute myocardial infarction | 8 | 0.1 | 73.7 | 6.5 |
| Shock | 9 | 1.0 | 74.7 | 4.0 |
| Thromboembolism, pulmonary embolism | 10 | 0.8 | 75.5 | 3.9 |
| Cerebrovascular disorders | 11 | 1.3 | 76.8 | 3.2 |
| Sepsis | 12 | 1.0 | 77.8 | 2.6 |
| Both DIC[f] & blood transfusion | 13 | 1.0 | 78.8 | 2.6 |
| Acute renal failure | 14 | 0.7 | 79.5 | 2.5 |
| Hysterectomy | 15 | 0.6 | 80.1 | 1.0 |
| Sickle cell crisis | 16 | 0.1 | 80.2 | 1.0 |
| Heart failure during surgery or procedure | 17 | 0.1 | 80.3 | 0.9 |
| Severe anesthetic complications | 18 | 0.1 | 80.4 | 0.7 |
| Pulmonary edema/acute congestive heart failure | 19 | 0.2 | 80.6 | 0.7 |
| DIC without blood transfusion | 20 | 0.3 | 80.9 | 0.2 |
| Blood transfusion without DIC[f] | 21 | 0.7 | 81.6 | 0.2 |
| Severe pre-eclampsia, HELLP[f] syndrome, or eclampsia | 22 | 0.5 | 82.1 | 0.1 |

[a] Three of the 21 current SMM indicators were excluded from this analysis, while four recoded indicators were included.

[b] The order of the indicators is based on the hierarchical, iterative process to determine the ranking of indicators. The indicator with the highest signal-to-noise ratio was selected at each iteration. The selected indicator is mutually exclusive of the remaining indicators, regardless of whether the hospitalization included other co-occurring indicators.

[c] The hierarchical population attributable fraction (hPAF) value is displayed for an indicator after previously selected indicators were made mutually exclusive. It is calculated controlling for year, payer, age, race, hospital region, and hospital location/teaching status and for all remaining (in hierarchical order) indicators.

[d] The cumulative hPAF is calculated as the sum of hPAF for the individual indicator and those above in ranking.

[e] Signal-to-noise ratio (SNR) is the selection criterion used to select the indicator that would maximize the ratio of the expected number of SMM cases detected (signal) to the increase in the standard deviation of the cumulative sum of indicators (noise).

[f] Other abbreviations: DIC: disseminated intravascular coagulation. HELLP syndrome: hemolysis, elevated liver enzymes, and a low platelet count.

Table 2 shows the order of the indicators established by our iterative process with hPAF and SNR values, excluding the three current indicators with measurement issues and adding four recoded indicators (n = 22 indicators). The indicators with the three highest hPAFs, mechanical ventilation (29.4%), restoration of cardiac rhythm (25.2%), and cardiac arrest (14.4%), were ranked third, first and second, respectively. Cardiac arrest had the highest SNR (69.4).

Among all delivery hospitalizations, the 22 SMM indicators identified 82% of in-hospital mortality. The top three ranked indicators (restoration of cardiac rhythm, cardiac arrest, mechanical ventilation) identified approximately 69% of in-hospital mortality. The top 15 ranked SMM indicators accounted for 80% of in-hospital mortality, while the bottom seven indicators together only identified an additional 2% of in-hospital mortality. Therefore, the top 15 indicators were combined into a ranked index for further analysis.

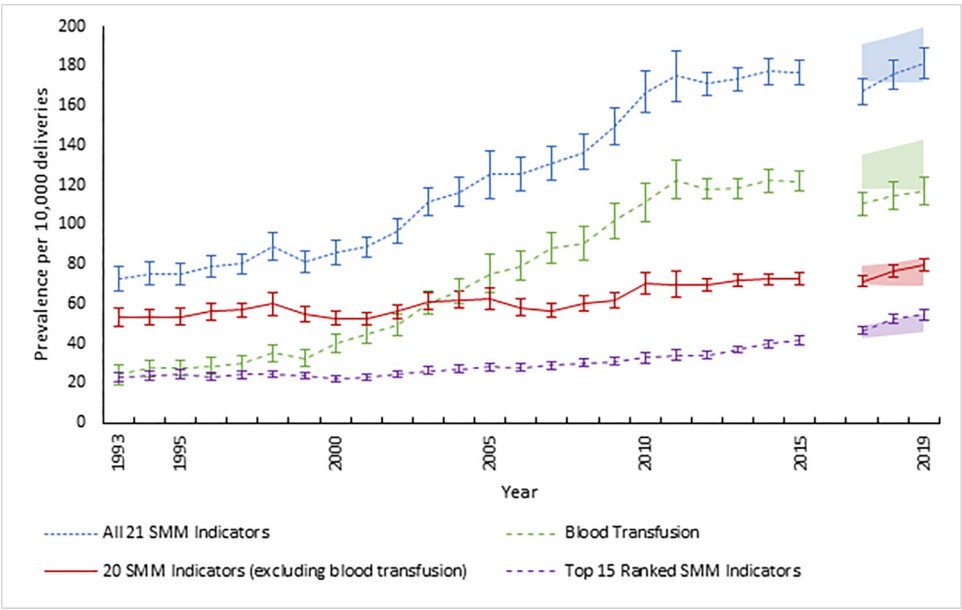

**Fig 1. Prevalence of severe maternal morbidity (SMM) per 10,000 delivery hospitalizations for (1) all 21 SMM indicators, (2) 20 SMM indicators (excluding blood transfusion), (3) blood transfusion alone, and (4) the top 15 ranked SMM indicators that identify the most in-hospital mortality, NIS, 1993–2019.** The blue, dotted line represents the 21 current SMM indicators developed by the Centers for Disease Control and Prevention (CDC). The green, dashed line represents the current indicator for blood transfusion alone. The red, solid line represents 20 SMM indicators (excluding blood transfusion). The purple dashed line represents the top 15 ranked SMM indicators that identify the most in-hospital mortality. Shaded areas represent the 95% confidence intervals for the regression-based SMM prevalence predictions, using 2010–2015 (ICD-9 coding) data, in 2017–2019 (ICD-10 coding). Interval colors correspond to the respective line colors. Data from the last quarter of 2015 and for the year 2016 were omitted because of changes in coding practices during the first year after ICD-10 coding implementation.

From 1993–2015, the overall SMM prevalence using the combined 21 current SMM indicators increased 143%, from 72.9 to 177.1 per 10,000 delivery hospitalizations (Fig 1). During this time, the prevalence of blood transfusion increased by almost 400%, driving a large portion of the increase in prevalence. The SMM prevalence of the 20 current SMM indicators, excluding blood transfusion, increased by 37% from 1993–2015. By contrast, the prevalence of the top 15 ranked SMM indicators increased by 79%, from 23.4 to 41.8 per 10,000 delivery hospitalizations from 1993–2015.

The observed cumulative SMM prevalence estimates are within the predicted 95% confidence intervals in 2017 and 2019 among delivery hospitalizations, but fell slightly outside in 2018 (Table 3, Fig 1). Eleven of the 15 indicators had observed prevalences within the predicted 95% confidence interval in 2017 and 2019, while nine did in 2018. In 2019, the estimated SMM prevalence using the top 15 ranked indicators that identified the most in-hospital mortality was 55 per 10,000 delivery hospitalizations (approximately SMM 21,800 [range 20,800 to 22,900] cases per year assuming 4 million delivery hospitalizations per year nationally).

## Discussion

Among approximately 88 million delivery hospitalizations, 15 SMM indicators identified 80% of in-hospital mortality in 1993–2015. The overall prevalence of this 15-indicator ranked SMM index increased by approximately 80% from 1993–2015. In contrast, while the prevalence of the current 20 SMM indicators, excluding blood transfusion, only increased 37% from 1993–2015. Using predicted estimates for 2017–2019, the prevalence of this 15-indicator ranked

**Table 3. Observed and predicted prevalences for years 2017–2019 per 10,000 deliveries of the 15 highest ranked severe maternal morbidity (SMM) indicators for in-hospital mortality, Nationwide Inpatient Sample (weighted sample = 11,001,554, unweighted sample = 2,200,312).**

| | | 2017 | | | 2018 | | | 2019 | | |
|---|---|---|---|---|---|---|---|---|---|---|
| Rank | Indicator | Observed[a] | Predicted (95% CI)[b] | Difference[c] | Observed | Predicted (95% CI)[b] | Difference[c] | Observed | Predicted (95% CI)[b] | Difference[c] |
| 1 | Restoration of cardiac rhythm | 0.85 | 0.65 (0.46, 0.91) | 0.20 | 0.95 | 0.63 (0.41, 0.95) | 0.32 | 0.99 | 0.61 (0.38, 0.99) | 0.38 |
| 2 | Cardiac arrest | 0.93 | 0.88 (0.63, 1.23) | 0.05 | 1.21 | 0.91 (0.61, 1.36) | 0.30 | 1.05 | 0.93 (0.58, 1.50) | 0.12 |
| 3 | Mechanical ventilation | 4.85 | 5.05 (4.41, 5.77) | −0.20 | 5.21 | 4.98 (4.24, 5.86) | 0.23 | 4.74 | 4.92 (4.07, 5.95) | -0.18 |
| 4 | Tracheostomy | 0.18 | 0.26 (0.14, 0.47) | −0.08 | 0.23 | 0.26 (0.12, 0.54) | −0.03 | 0.24 | 0.26 (0.11, 0.61) | -0.02 |
| 5 | Amniotic fluid embolism | 0.62 | 0.36 (0.23, 0.55) | 0.26 | 0.62 | 0.34 (0.21, 0.58) | 0.28 | 0.64 | 0.33 (0.18, 0.60) | 0.31 |
| 6 | Aneurysm | 0.34 | 0.47 (0.27, 0.81) | −0.13 | 0.41 | 0.55 (0.29, 1.06) | −0.14 | 0.38 | 0.64 (0.30, 1.39) | -0.26 |
| 7 | Acute respiratory distress syndrome | 9.21 | 8.59 (7.66, 9.62) | 0.62 | 10.25 | 8.78 (7.65, 10.08) | 1.47 | 10.14 | 8.97 (7.62, 10.56) | 1.17 |
| 8 | Acute myocardial infarction | 0.31 | 0.17 (0.09, 0.32) | 0.14 | 0.37 | 0.16 (0.08, 0.34) | 0.21 | 0.49 | 0.16 (0.07, 0.37) | 0.33 |
| 9 | Shock | 5.55 | 6.15 (5.33, 7.09) | −0.60 | 6.84 | 6.88 (5.79, 8.18) | −0.04 | 6.92 | 7.70 (6.29, 9.43) | -0.78 |
| 10 | Thromboembolism, pulmonary embolism | 3.02 | 2.26 (1.84, 2.77) | 0.76 | 3.62 | 2.31 (1.80, 2.96) | 1.31 | 2.99 | 2.37 (1.77, 3.16) | 0.62 |
| 11 | Cerebrovascular disorders | 2.82 | 3.77 (3.21, 4.43) | −0.95 | 3.40 | 3.81 (3.14, 4.62) | −0.41 | 3.50 | 3.84 (3.06, 4.82) | -0.34 |
| 12 | Both DIC & blood transfusion | 4.90 | 6.45 (5.71, 7.29) | −1.55 | 4.81 | 6.48 (5.59, 7.51) | −1.67 | 4.66 | 6.50 (5.46, 7.74) | -1.84 |
| 13 | Acute renal failure | 11.47 | 10.29 (9.12, 11.61) | 1.18 | 13.04 | 11.14 (9.63, 12.89) | 1.90 | 15.27 | 12.05 (10.15, 14.31) | 3.22 |
| 14 | Sepsis | 9.37 | 9.45 (8.20, 10.90) | −0.08 | 10.55 | 10.80 (9.09, 12.84) | −0.25 | 11.14 | 12.33 (10.06, 15.10) | -1.19 |
| 15 | Hysterectomy | 11.65 | 12.63 (11.39, 14.01) | −0.98 | 12.67 | 13.28 (11.71, 15.07) | −0.61 | 12.53 | 13.97 (12.03, 16.23) | -1.44 |
| | Top 15 ranked SMM indicators (cumulative) | 46.62 | 45.83 (43.02, 48.81) | 0.79 | 52.40 | 48.20 (44.60, 52.08) | 4.20 | 54.63 | 50.69 (46.24, 55.57) | 3.94 |

[a] According to the hierarchical population attributable fraction approach in the ranked SMM index for in-hospital mortality.

[b] Predictions and 95% confidence intervals were estimated based on logistic regression models using observed data from 2010 to 2015, accounting for the complex sample design of the Healthcare Cost and Utilization Project National Inpatient Sample.

[c] Difference calculated by subtraction: Observed–Predicted. A positive number indicates the observed prevalence was greater than the predicted prevalence. A negative number indicates the predicted prevalence was greater than the observed. Rounding occurred.

SMM index is 54.6 per 10,000 delivery hospitalizations or approximately 22,000 SMM cases per year.

A major strength of this study is the use of NIS, a large, representative national all-payer database widely used in the US to provide generalizable estimates and trends. NIS provided a unique opportunity to determine which SMM indicators, though rare outcomes, identified the most in-hospital mortality at delivery hospitalization. Our study has several limitations. First, the SMM indicators include both diagnosis and procedure-based indicators from administrative data, for which coding rules change over time. Procedure-based indicators may be a proxy for multiple diagnosis-based indicators. For example, ventilation may be an indirect indicator of acute respiratory distress syndrome, pulmonary edema, and/or pulmonary embolism.

Second, we used pre-defined conditions or procedures to ascertain SMM indicators from the hospital discharge records. The indicators identified 82% of in-hospital mortality, therefore approximately 18% is attributable to conditions not identified by these analyses. Conditions not included as indicators of severe morbidity that identify in-hospital mortality include those that clinicians encounter in the Intensive Care Unit (ICU; e.g., motor vehicle crashes, poisoning, suicides, malignancies). The NIS does not include ICU admission status for all reporting states [16]. Additionally, the SMM indicators do not capture information on comorbidities such as mental health conditions or substance use disorders which may also contribute to in-hospital mortality. Third, our study only examines in-hospital mortality at delivery hospitalization and does not look at antepartum or postpartum hospitalizations. This study does not capture the full spectrum of pregnancy related deaths that occur during pregnancy or up to one year postpartum. Fourth, we did not incorporate length of stay (LOS) in our algorithm. The SMM indicators with and without LOS considerations demonstrated excellent reliability (kappa statistic = 0.927) in a study from California [21]. Hence, its inclusion may add methodological complexity and may not contribute to meaningful interpretation of the association between SMM and in-hospital mortality. Fifth, the quality of documentation by physicians, coders' training, and ongoing initiatives may affect coding accuracy [22]. Several validity studies reported high to moderate specificity but low sensitivity for some ICD-9 codes for SMM [11–13]. Although limited studies on the validity of ICD-10 codes are available, the sensitivity of these codes may be lower than for ICD-9 codes [13], though it may vary by indicator. Sixth, we were limited in examining pulmonary edema and acute congestive heart failure separately due to interchangeable coding practices for pulmonary edema and heart failure [23]. Pulmonary edema is labeled as a left ventricular failure in the coding manual and described as a left ventricular failure or heart failure in the medical chart [23], though ambiguity of medical diagnoses like this are inherent when using administrative data sets for analyses rather than individual medical records. Finally, we combined the top 15 SMM indicators into a ranked index for further analysis because these indicators identified 80% of in-hospital mortality and the remaining indicators only identified 2%; however, this threshold of 15 was arbitrary.

Our study offers an index for SMM population-level surveillance. While there are limitations to using administrative data to identify SMM, the ranked SMM index provides a measurement that is associated with the most severe outcome, in-hospital mortality, and can be used as a marker for maternal health.

The top three ranked indicators (restoration of cardiac rhythm, cardiac arrest, mechanical ventilation), all markers of acute cardiac and respiratory compromise, identified approximately 69% of in-hospital mortality. While cardiac arrest may result from a variety of cardiovascular and non-cardiovascular conditions, cardiovascular conditions are among the leading causes of maternal death [24]. This highlights the importance of rapid response teams, checklists, huddles, and debriefs to ensure successful and prompt resuscitation [25, 26].

The most prevalent indicators were the recoded severe type of hypertension of pregnancy indicator, the current blood transfusion indicator, and the recoded blood transfusion without DIC indicator; however, none of these indicators ranked among the top 15 indicators. Hypertensive disorders in pregnancy and postpartum hemorrhage are significant contributors to maternal morbidity and mortality and have increased in prevalence over the last two decades [24, 27]. In our analysis, we recoded the eclampsia indicator to include severe pre-eclampsia and HELLP syndrome in order to capture a broader spectrum of hypertensive disorders in pregnancy. While the severe pre-eclampsia, HELLP syndrome, or eclampsia indicator was last on our mutually exclusive, ranked SMM index, it is possible that an SMM code indicating a consequence of severe pre-eclampsia, HELLP syndrome, or eclampsia that is more strongly associated with in-hospital mortality (e.g., a puerperal cerebrovascular disorder; mechanical

ventilation) would also be present. This would produce a lower hierarchical ranking of this indicator individually. The current blood transfusion indicator and the recoded blood transfusion without DIC indicator were the second and third most prevalent indicators. Obstetric transfusions are primarily associated with postpartum hemorrhage, and we sought to capture severe cases of postpartum hemorrhage by recoding the blood transfusion and DIC indicators. The NIS data do not quantify transfusion of blood products in units which might serve as a proxy for severity [28]. However, hospital discharge records with severe blood loss resulting in hemorrhagic shock are expected to have ICD codes for blood transfusion and multiple organ failure, including DIC. Thus, most hospitalizations with severe hemorrhage in our study should be captured by indicators of organ failure. The recoded indicator for both DIC and blood transfusion was among the 15 top ranked indicators. By making indicators mutually exclusive during the ranking process, we identified how many *additional SMM deaths* an indicator would capture, accounting for indicators already ranked. This process does not account for the true burden of the condition which informs population-based initiatives for prevention through organizations such as perinatal quality collaboratives [29].

Our methodology allowed the ranking of rare indicators with high in-hospital mortality and requires further testing. The strength and quality of evidence generated by its use could provide the basis for further refinement of SMM indicators, increasing potential applicability. Validation studies of the indicators used for the ranked SMM index are needed and may inform index utility.

Our results have important public health and surveillance implications. While this ranked SMM index results in a lower rate and total number of SMM cases compared with estimates of SMM using the combined 21 current indicators, it directly addresses the criticism of the combined 21 current SMM indicators: a lack of specificity for disease severity [11, 21]. While "severe" is difficult to define, linking diagnosis and procedure codes to in-hospital mortality underscores that selected conditions and events precede the most severe event, death. However, a ranking of indicators for the purpose of determining how many deaths could be prevented by sequentially eliminating indicators would need to use a different definition of PAF than we used (e.g., Benichou 2007) [30]; in particular, our hPAF is not appropriate for this purpose.

The ranked SMM index is for surveillance at the population level and was not developed for individual facility reviews [8]. Criteria for facility review of SMM are outlined by clinical organizations such as the American College of Obstetricians and Gynecologists and the Society for Maternal and Fetal Medicine [31]. The indicators included in the ranked SMM index could facilitate further refinement of SMM indicators for other uses (e.g., facility-based identification and review of SMM) and improve overall utility.

The ranked SMM index is unsuitable for comparison of individual health care institutions for multiple reasons. Health care institutions have access to the medical record and detailed and reliable data on comorbidities and severity, unavailable in the NIS, yielding more accurate, informative, and comparable case results. Comorbid conditions are strong predictors of overall SMM [32], but comorbid conditions are under-reported in hospital discharge data [33], with limited studies of reliability of ICD codes among hospitals [34], especially among delivery hospitalizations. The NIS data lack information on prenatal care and antepartum risk factors that are important determinants of SMM [35]. These factors are not considered in this ranked SMM index to adjust for differences in case-mix among institutions. Facilities with higher levels of maternal care are expected to have more high-risk deliveries and may have higher SMM rates compared with lower-level facilities [32]. Our approach of creating mutually exclusive indicators through iterative ranking may not be appropriate for care management goals in individual health care institutions. For instance, the risk of in-hospital death and thus, the

condition severity and complexity of clinical management, increases with the growing number of SMM indicators [36]. Clinical reviews allow identification of preventable causes of SMM including determinants of clinical practices associated with SMM [31]. Only clinical reviews can guide design of hospital-level interventions and quality improvement initiatives to reduce SMM. A recent study of ICD-10 coding for blood transfusion, DIC, and acute renal failure confirmed wide variability of code sensitivity among hospitals in administrative discharge data and cautioned against using this type of data to compare inter-hospital safety and quality improvement initiatives at the facility level [13].

## Conclusions

This study offers an index for reporting severe conditions using 15 indicators that identify 4 in 5 in-hospital deaths at delivery hospitalization. These indicators, when considered together, yield an index for population-level SMM surveillance based on in-hospital mortality. SMM surveillance is a promising method to further the understanding of maternity care and ultimately improve it [37]. Surveillance quality can be improved by reviewing medical records and corresponding ICD-10 codes to determine validity. Since a substantial proportion of pregnancy-related deaths and serious pregnancy-related complications occur outside of the delivery hospitalization, additional indices for antenatal and postpartum periods are needed [24]. Our findings underscore the need for continued testing of SMM indicators to refine metrics that provide simplified measurement of severe obstetric complications at the population level.

## Supporting information

**S1 Appendix. The signal to noise ratio: Formula for calculation.**
(DOCX)

**S1 Table. Characteristics of study population, Nationwide Inpatient Sample, 1993–2015 (weighted sample = 87,864,173, unweighted sample = 18,198,934).**
(DOCX)

## Acknowledgments

**Disclaimer:** The findings and conclusions in this paper are those of the authors and do not necessarily represent the official position of the Centers for Disease Control and Prevention.

## Author Contributions

**Conceptualization:** Elena V. Kuklina, Alexander C. Ewing, Glen A. Satten, William M. Callaghan, David A. Goodman, Cynthia D. Ferre, Jean Y. Ko, Romeo R. Galang, Charlan D. Kroelinger.

**Data curation:** Elena V. Kuklina, Alexander C. Ewing.

**Formal analysis:** Elena V. Kuklina, Alexander C. Ewing, Glen A. Satten, Charlan D. Kroelinger.

**Methodology:** Elena V. Kuklina, Alexander C. Ewing, Glen A. Satten, William M. Callaghan, Jean Y. Ko, Charlan D. Kroelinger.

**Project administration:** Charlan D. Kroelinger.

**Resources:** Charlan D. Kroelinger.

**Supervision:** Glen A. Satten, William M. Callaghan, David A. Goodman, Cynthia D. Ferre, Jean Y. Ko, Charlan D. Kroelinger.

**Validation:** Lindsay S. Womack.

**Writing – original draft:** Elena V. Kuklina, Alexander C. Ewing, Glen A. Satten, William M. Callaghan, David A. Goodman, Cynthia D. Ferre, Jean Y. Ko, Romeo R. Galang, Charlan D. Kroelinger.

**Writing – review & editing:** Glen A. Satten, Cynthia D. Ferre, Jean Y. Ko, Lindsay S. Womack, Romeo R. Galang, Charlan D. Kroelinger.

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
