## [Decision Letter · Decision Letter 0]

9 May 2023

PONE-D-23-10565Ranked Severe Maternal Morbidity Index for Population-Level Surveillance at Delivery HospitalizationPLOS ONE

Dear Dr. Womack,

Thank you for submitting your manuscript to PLOS ONE. After careful consideration, we feel that it has merit but does not fully meet PLOS ONE’s publication criteria as it currently stands. Therefore, we invite you to submit a revised version of the manuscript that addresses the points raised during the review process.

We look forward to receiving your revised manuscript.

Kind regards,

José Paulo de Siqueira Guida, PhD

Academic Editor

PLOS ONE

Journal Requirements:

Additional Editor Comments:

Dear Authors,

thanks for the submission of the manuscript to PLOS ONE. Please find attached some minor revisions suggested by revieweres.

Reviewer #1

This is an important and well-written study that I believe adds substantial information to the study of severe maternal morbidity.

I have just a few observations and suggestions that require only minor revisions.

1. The study design/source should be presented in the title.

2. The number of hospitalizations should be presented in the abstract.

3. I suggest that the total number of deaths, in addition to the rate, be presented in the results section.

That is all. Congratulations on your work.

Reviewer #2

This manuscript uses data from the NIS to estimate the relative contribution of SMM indicators to in-hospital mortality. The approach expands the findings from a Canadian study cited by the authors. The inclusion of the signal-to-noise ratio is novel and useful for interpreting the PAF for factors with high prevalence but low mortality. The text is well written and the conclusions are supported by the results with appropriate acknowledgement of the limitations of the study.

1. Introduction. Suggest reporting the range of sensitivity for the 3 indicators reported to have low sensitivity - DIC, transfusion, eclampsia

2. Is there evidence that coding for HELLP and severe preeclampsia is any more valid than eclampsia coding? This seems relevant given that this combined group of hypertensive conditions had the highest prevalence but a low contribution to mortality. Did the authors consider a sensitivity analysis using eclampsia alone?

3. It might be helpful to include a sentence helping readers interpret the SNR - are high or low values better?

4. The methods should include the information reported in footnote 'c' for Table 2 - that hPAFs controlled for year, payer, race, region, hospital teaching status. How were these covariates selected? It also seems that there should be a descriptive table reporting the distribution of these characteristics for the study population. Race is often missing in administrative data. Was this also the case for this study and if so, how was it addressed? Also, consistent with recommended best practices (https://jamanetwork.com/journals/jama/article-abstract/196632), please note how race data are collected in NIS and why race was included in the analysis.

Reviewers' comments:

Reviewer's Responses to Questions

**Comments to the Author**

1. Is the manuscript technically sound, and do the data support the conclusions?

Reviewer #1: Yes

Reviewer #2: Yes

2. Has the statistical analysis been performed appropriately and rigorously? 

Reviewer #1: Yes

Reviewer #2: Yes

3. Have the authors made all data underlying the findings in their manuscript fully available?

Reviewer #1: No

Reviewer #2: Yes

4. Is the manuscript presented in an intelligible fashion and written in standard English?

Reviewer #1: Yes

Reviewer #2: Yes

5. Review Comments to the Author

Reviewer #1: This is an important and well-written study that I believe adds substantial information to the study of severe maternal morbidity.

I have just a few observations and suggestions that require only minor revisions.

1. The study design/source should be presented in the title.

2. The number of hospitalizations should be presented in the abstract.

3. I suggest that the total number of deaths, in addition to the rate, be presented in the results section.

That is all. Congratulations on your work.

Reviewer #2: This manuscript uses data from the NIS to estimate the relative contribution of SMM indicators to in-hospital mortality. The approach expands the findings from a Canadian study cited by the authors. The inclusion of the signal-to-noise ratio is novel and useful for interpreting the PAF for factors with high prevalence but low mortality. The text is well written and the conclusions are supported by the results with appropriate acknowledgement of the limitations of the study.

1. Introduction. Suggest reporting the range of sensitivity for the 3 indicators reported to have low sensitivity - DIC, transfusion, eclampsia

2. Is there evidence that coding for HELLP and severe preeclampsia is any more valid than eclampsia coding? This seems relevant given that this combined group of hypertensive conditions had the highest prevalence but a low contribution to mortality. Did the authors consider a sensitivity analysis using eclampsia alone?

3. It might be helpful to include a sentence helping readers interpret the SNR - are high or low values better?

4. The methods should include the information reported in footnote 'c' for Table 2 - that hPAFs controlled for year, payer, race, region, hospital teaching status. How were these covariates selected? It also seems that there should be a descriptive table reporting the distribution of these characteristics for the study population. Race is often missing in administrative data. Was this also the case for this study and if so, how was it addressed? Also, consistent with recommended best practices (https://jamanetwork.com/journals/jama/article-abstract/196632), please note how race data are collected in NIS and why race was included in the analysis.

6. PLOS authors have the option to publish the peer review history of their article (what does this mean?). If published, this will include your full peer review and any attached files.

Reviewer #1: No

Reviewer #2: No

---

## [Author Response · Author response to Decision Letter 0]

16 May 2023

RESPONSE TO JOURNAL REQUIREMENTS:

JOURNAL REQUIREMENTS (SEE BELOW):

R: Thank you for providing the style templates, we have edited the manuscript to meet the style requirements.

R: Thank you for bringing this to our attention. We updated the Data Availability Statement as indicated above in the body of our letter.

R: We added captions for our Supporting Information files at the end of the manuscript. We also updated the in-text citations. Thank you for providing the guidelines.

R: We reviewed our reference list to ensure it is complete and correct. We made minor formatting edits to align with the journal’s reference style. To our knowledge, we have not cited any papers that were retracted. We updated the citation for reference #19 (Rockhill et al.) to include that an erratum was published and provided the reference for the erratum.

The updated citation in our paper is now listed as: Rockhill B, Newman B, Weinberg C. Use and misuse of population attributable fractions. [published correction appears in Am J Public Health. 2008 Dec;98(12):2119]. Am J Public Health. 1998;88(1):15-19. doi:10.2105/ajph.88.1.15

 

RESPONSE TO REVIEWERS:

REVIEWERS' COMMENTS (SEE BELOW):

Reviewer #1:

C1: This is an important and well-written study that I believe adds substantial information to the study of severe maternal morbidity. I have just a few observations and suggestions that require only minor revisions.

R1: We thank the reviewer for their positive comments, and we appreciate their suggestions to improve the manuscript.

C2: The study design/source should be presented in the title.

R2: We appreciate the reviewer’s and we have edited the title to include the data source: Ranked severe maternal morbidity index for population-level surveillance at delivery hospitalization based on hospital discharge data

C3: The number of hospitalizations should be presented in the abstract.

R3: Thank you for this suggestion. We added a sentence in the Results section of the Abstract that provides the total number of delivery hospitalizations (page 2, lines 34–35 of the revised manuscript with track changes).

C4: I suggest that the total number of deaths, in addition to the rate, be presented in the results section.

R4: We appreciate this comment and added the total number of deaths to the second sentence of the Results section (page 13, line 195 of the revised manuscript with track changes).

Reviewer #2:

C1: This manuscript uses data from the NIS to estimate the relative contribution of SMM indicators to in-hospital mortality. The approach expands the findings from a Canadian study cited by the authors. The inclusion of the signal-to-noise ratio is novel and useful for interpreting the PAF for factors with high prevalence but low mortality. The text is well written and the conclusions are supported by the results with appropriate acknowledgement of the limitations of the study.

R1: We appreciate the reviewer’s comments and suggestions in their review.

C2: Introduction. Suggest reporting the range of sensitivity for the 3 indicators reported to have low sensitivity - DIC, transfusion, eclampsia

R2: Thank you for this suggestion. We added the previously reported sensitivity from the Friedman et al. study for blood transfusion and DIC in the Introduction (page 4, line 64 of the revised manuscript with track changes). We did not include the sensitivities for the Main et al. study (reference #11) because that study looked at the overall sensitivity of the combined SMM indicators when blood transfusion is included and excluded. It found that the overall sensitivity improved when blood transfusion was removed from the composite measure. We think it is important to cite and shows that the sensitivity of blood transfusion is low; however, we cannot report the exact sensitivity for blood transfusion. We also updated the reference citations and edited the text to report the positive predictive value for eclampsia in the Introduction (page 4, lines 66–67). The study we cited (Geller et al.) reported PPV and not sensitivity. However, the relatively low PPV of 41.7% still emphasizes this indicator may not truly capture severe events. We removed the citation to the Coolman et al. study (previous reference #14) because it did not have a PPV or sensitivity for eclampsia that we could report.

C3: Is there evidence that coding for HELLP and severe preeclampsia is any more valid than eclampsia coding? This seems relevant given that this combined group of hypertensive conditions had the highest prevalence but a low contribution to mortality. Did the authors consider a sensitivity analysis using eclampsia alone?

R3: Thank you for this comment. We decided to include severe pre-eclampsia, HELLP syndrome (hemolysis, elevated liver enzymes, and low platelet count), and eclampsia to be consistent with the Canadian SMM index. Regarding the validity of the eclampsia indicator compared with the combined severe pre-eclampsia, HELLP syndrome, and eclampsia indicator, a previous study conducted by Geller et al. reports that the PPV of severe preeclampsia (84.5%) is higher than eclampsia (41.7%); they did not assess the validity of HELLP syndrome. These findings suggest the combined indicator has a higher validity than eclampsia alone. Furthermore, in Table 1, we present the prevalence, in-hospital mortality, PAF, and SNR of both the eclampsia indicator and the combined severe pre-eclampsia, HELLP syndrome, and eclampsia indicator. The table shows that the PAF and SNR are higher for the combined indicator compared with eclampsia alone. For this reason, we decided to use the combined indicator in the ranking and did not conduct a sensitivity analysis using eclampsia alone.

C4: It might be helpful to include a sentence helping readers interpret the SNR - are high or low values better?

R4: Thank you for this suggestion to help readers interpret the SNR. We added the following sentence to the end of the first paragraph in the Statistical Analysis section: “A higher SNR value suggests that an indicator’s higher PAF is due to a higher mortality rate rather than a higher prevalence in the population.” (page 7, lines 130–132 of the revised manuscript with track changes).

C5: The methods should include the information reported in footnote 'c' for Table 2 - that hPAFs controlled for year, payer, race, region, hospital teaching status. How were these covariates selected? It also seems that there should be a descriptive table reporting the distribution of these characteristics for the study population. Race is often missing in administrative data. Was this also the case for this study and if so, how was it addressed? Also, consistent with recommended best practices (https://jamanetwork.com/journals/jama/article-abstract/196632), please note how race data are collected in NIS and why race was included in the analysis.

R5: Thank you for this comment. We edited the Statistical Analysis section in the Methods and added information on how the PAFs were calculated (page 7, lines 119–124 of the revised manuscript with track changes). This was information previously included in the footnotes of Table 1 and Table 2. We also removed the parenthetical comment in the Methods to the footnote in Table 1 (page 7, line 119 of the revised manuscript with track changes), since this information is now included in the text. Because of this edit, we moved Table 1 to appear in the Results section.

We selected the covariates (including race) for the regression model a priori, based on the literature and clinical significance. We added this rationale to the Methods section (page 7, lines 125–126 of the revised manuscript with track changes). In this analysis, we are not reporting specific racial/ethnic or hospital location results. For this reason, we decided to treat the “missing/unknown” values as a separate category. We added this information to the Methods sections (page 7, lines 124–125 of the revised manuscript with track changes), and we added a reference to the HCUP NIS website that describes how race data are collected in the NIS (page 7, line 123 of the revised manuscript with track changes).

Finally, we added a supplemental table to describe study population and reference the supplemental table in first sentence of the Results section (page 13, line 194 of the revised manuscript with track changes).

END

---

## [Decision Letter · Decision Letter 1]

26 Oct 2023

Ranked severe maternal morbidity index for population-level surveillance at delivery hospitalization based on hospital discharge data

PONE-D-23-10565R1

Dear Dr. Womack,

We’re pleased to inform you that your manuscript has been judged scientifically suitable for publication and will be formally accepted for publication once it meets all outstanding technical requirements.

Kind regards,

Simone Garzon

Academic Editor

PLOS ONE

Additional Editor Comments (optional):

Reviewers' comments:

Reviewer's Responses to Questions

**Comments to the Author**

1. If the authors have adequately addressed your comments raised in a previous round of review and you feel that this manuscript is now acceptable for publication, you may indicate that here to bypass the “Comments to the Author” section, enter your conflict of interest statement in the “Confidential to Editor” section, and submit your "Accept" recommendation.

Reviewer #1: All comments have been addressed

Reviewer #2: All comments have been addressed

2. Is the manuscript technically sound, and do the data support the conclusions?

Reviewer #1: Yes

Reviewer #2: Yes

3. Has the statistical analysis been performed appropriately and rigorously? 

Reviewer #1: Yes

Reviewer #2: Yes

4. Have the authors made all data underlying the findings in their manuscript fully available?

Reviewer #1: Yes

Reviewer #2: Yes

5. Is the manuscript presented in an intelligible fashion and written in standard English?

Reviewer #1: Yes

Reviewer #2: Yes

6. Review Comments to the Author

Reviewer #1: Dear authors:

I read and greatly appreciated the responses to my comments as a reviewer of your interesting article and I was pleased to see that all my suggestions were incorporated into the text. As I previously highlighted, it is an important article that makes a substantial contribution to the study of severe maternal morbidity. Therefore, I recommend its approval in this version.

Reviewer #2: (No Response)

7. PLOS authors have the option to publish the peer review history of their article (what does this mean?). If published, this will include your full peer review and any attached files.

Reviewer #1: **Yes: **Melania Maria Ramos de Amorim

Reviewer #2: No

---

## [Editor Report · Acceptance letter]

31 Oct 2023

PONE-D-23-10565R1 

Ranked severe maternal morbidity index for population-level surveillance at delivery hospitalization based on hospital discharge data 

Dear Dr. Womack:

I'm pleased to inform you that your manuscript has been deemed suitable for publication in PLOS ONE. Congratulations! Your manuscript is now with our production department. 

Kind regards, 

on behalf of

Dr. Simone Garzon 

Academic Editor

PLOS ONE